# The Utility of Three-Dimensional Printing in Physician-Modified Stent Grafts for Aortic Lesions Repair

**DOI:** 10.3390/jcm13102977

**Published:** 2024-05-18

**Authors:** Wiktoria Antonina Zasada, Hubert Stępak, Magdalena Węglewska, Łukasz Świątek, Jerzy Kluba, Zbigniew Krasiński

**Affiliations:** 1Students’ Research Group of Vascular Surgery, Poznan University of Medical Sciences, Rokietnicka 7 Street, 60-608 Poznań, Poland; 2Department of Vascular and Endovascular Surgery, Angiology and Phlebology, Poznan University of Medical Sciences, Długa Street, 61-848 Poznan, Poland

**Keywords:** 3D printing, AAA, FEVAR, TEVAR, PMSG

## Abstract

**Background**: Three-dimensional (3D) printing is becoming increasingly popular around the world not only in engineering but also in the medical industry. This trend is visible, especially in aortic modeling for both training and treatment purposes. As a result of advancements in 3D technology, patients can be offered personalized treatment of aortic lesions via physician-modified stent grafts (PMSG), which can be tailored to the specific vascular conditions of the patient. The objective of this systematic review was to investigate the utility of 3D printing in PMSG in aortic lesion repair by examining procedure time and complications. **Methods**: The systematic review has been performed using the PRISMA 2020 Checklist and PRISMA 2020 flow diagram and following the Cochrane Handbook. The systematic review has been registered in the International Prospective Register of Systematic Reviews: CRD42024526950. **Results**: Five studies with a total number of 172 patients were included in the final review. The mean operation time was 249.95± 70.03 min, and the mean modification time was 65.38 ± 10.59 min. The analysis of the results indicated I^2^ of 99% and 100% indicating high heterogeneity among studies. The bias assessment indicated the moderate quality of the included research. **Conclusions**: The noticeable variance in the reviewed studies’ results marks the need for larger randomized trials as clinical results of 3D printing in PMSG have great potential for patients with aortic lesions in both elective and urgent procedures.

## 1. Introduction

Abdominal aortic aneurysm (AAA) is characterized by a dilation of the aorta in the abdominal region, exceeding 1.5 times the standard dimensions, predominantly situated beneath the renal arteries and at the bifurcation of the iliac arteries [1]. The prevalence of AAA ranges from 1.3% to 12.5% within the general population, making it a noteworthy health concern [2].

The etiology of AAA involves complex atherosclerotic processes and inflammatory mechanisms, leading to elevated synthesis of cytokines and proteinases [1,3]. These biochemical formations result in the degradation of collagen and elastin within the aortic wall, leading to aneurysm formation [4].

The possibility of AAA rupture is strongly correlated with its size, with a diameter greater than 5.5 cm for men and 5.0 cm for women warranting elective intervention, due to the increased risk of rupture [5].

Elevated risks of rupture in AAA are correlated with a larger aneurysm diameter; for AAAs of 4.0–4.9 cm, the risk is less than 5% per year, while for AAAs with a diameter larger than 7 cm, the risk can rise to 30% per year [5]. Another risk factor for rupture is a higher level of high-sensitivity C-reactive protein (hs-CRP), while a higher level of high-density lipoprotein cholesterol (HDL-C) is inversely associated with the risk of AAA rupture [6,7]. Notwithstanding, the Society for Endovascular Surgery suggests annual follow-up for patients with an AAA of 4.0–4.9 cm in diameter [8].

When discussing life-threatening conditions in the aorta, it is important to mention aortic dissection. This is a hematoma within the aortic wall that causes a penetrating detachment of the medial layer, resulting in the formation of a false lumen. This can lead to aortic rupture or re-entry into the true lumen of the aorta [9].

Research indicates that the adoption of Endovascular Aneurysm Repair (EVAR) offers superior outcomes compared to Open Repair (OR) [5]. Noteworthy advantages associated with EVAR include reduced perioperative mortality rates, lower complication rates, fewer needs for blood transfusions, shortened hospitalization durations, improved early-stage results, and better mental health [10]. Although decreased 30-day perioperative mortality is observed after EVAR, the OR shows significantly better results in midterm (2–6 years). Long-term mortality rates do not differ between OR and EVAR (>6 years after repair) [11].

When treating aneurysms positioned distally to the ligamentum arteriosum within the thoracic aorta, thoracic endovascular aortic repair (TEVAR) becomes an available and safer option due to lower mortality rates [12]. Studies have demonstrated the usefulness of branched endovascular aortic aneurysm repair (BEVAR) in the treatment of acute kidney injury caused by vascular defects [13].

Potential issues such as endograft migration and detection of endoleak, are observed when using stent grafts [13]. To address these concerns, further stenting and coiling may be employed, potentially resulting in a stiffer aortic wall [13]. The heightened stiffness of the stent graft, compared to the native aorta, has the capacity to alter the hemodynamic flow, leading to reduced coronary perfusion attributed to the Windkessel effect [13]. Postoperatively, the patients may experience symptoms such as hypertension and chest pain [13]. The increased load on the left ventricle may lead to left ventricular hypertrophy [13]. Notably, the risk of cardiovascular complications is more pronounced when deploying stents in close proximity to the heart and aortic valve [13,14].

In current medical practice, the generation of patient-specific models for AAA has become increasingly common. Advanced imaging technologies such as computed tomography (CT) scans and magnetic resonance imaging (MRI), coupled with 3D software tools such as SpaceClaim (ANSYS, Inc., Canonsburg, PA, USA), enable highly accurate design of anatomically correct AAA models fabricated from silicone-like materials, facilitating fenestrated endovascular aortic repair (FEVAR) [15]. These models serve as a platform for conducting parametric studies, facilitating a comprehensive analysis of hemodynamic parameters [16]. Moreover, they play a crucial role in the formulation of physician-modified stent grafts (PMSGs), customized to address the unique anatomical considerations of individual patients [17].

PMSGs are carefully prepared within a sterile environment by insertion into 3D-printed aortic models (3DAMs) [18]. Subsequently, the PMSG undergoes strategic rotation to position all fenestrations in regions free of structural obstructions within the stent graft [19]. The specific sites for the fenestrations are marked using a sterile pen, Followed by careful incisions made by the surgeon using a scalpel to accommodate vessels originating from the aneurysm. This approach ensures precise adaptation of the stent graft to the unique characteristics of the patient’s blood vessels. This procedure reduces the duration of modification—a critical consideration in patient management [20]. Depending on the location, the waiting period for custom-made devices (CMD) typically ranges from 12 to 15 weeks, a delay that may increase mortality rates by up to 4% [18].

There is also the possibility of preparing PMSG without using printed 3D models of the aorta. This is achievable through the utilization of a virtual 3D model of the vessel developed based on prior tomography of the affected segment [21].

Another method of preparing patient-specific stent grafts is laser in situ fenestration (LISF). It involves placing the stent graft within the patient’s body in the supplied segment of the aorta, and then creating openings, allowing blood flow to branches branching off from the vessel, using a laser. The LISF has been considered technically simpler and faster; however, PMSG has an advantage over LISF in its potential utility in treating aortic dissection [17].

When evaluating high-risk patients, EVAR proves to be more cost-effective than open repair (OR). However, studies have indicated that it involves escalated costs in low-risk patients [20]. Comprehensive research revealing the comparative costs between the standard procedures and those using 3D technology is not yet available, however, ongoing trials hold promise in providing this information [22].

Advancements in 3D technology contribute to the reduction of stent-graft preparation time, rates of reintervention, duration of hospital stays, and enhancement of long-term outcomes [18,19,20,23]. Reduction of preparation and modification time can contribute to more frequent use in urgent surgeries achieving precise personalized treatment. The aim of this systematic review is to examine the existing knowledge regarding the procedure of physician-modified stent-graft implantation using a 3D-printed model (Figure 1). This paper focuses on procedure characteristics such as operation and modification time, all related complications, and the final outcome of the treatment.

## 2. Materials and Methods

This systematic review was conducted following the guidelines outlined in the Cochrane Handbook for Systematic Reviews of Interventions and adhered to the Preferred Reporting Items for Systematic Reviews and Meta-Analyses (PRISMA) statement (Appendix A) [24,25]. The protocol for this review was registered in the International Prospective Register of Systematic Reviews (PROSPERO registration number: CRD42024526950).

### 2.1. Criteria for Considering Studies for This Review

#### 2.1.1. Types of Studies

Eligible study designs included randomized controlled trials (RCTs), quasi-experimental (QE) designs without randomization, retrospective analysis studies, and case series providing valuable insights into the patient-specific applications of three-dimensional printing.

#### 2.1.2. Types of Participants

This review considered studies involving participants: (i) both men and women, older than 18 years, representing all ethnic groups; (ii) with an aortic lesion eligible for FEVAR, BEVAR, or TEVAR procedures; (iii) under surveillance/screening.

#### 2.1.3. Types of Interventions

Included studies specifically targeted the usage of physician-modified stent grafts in FEVAR or TEVAR procedures using 3D-printed models. Studies were excluded if they failed to provide sufficient information on intervention content to assess the focus on physician-modified stent grafts.

#### 2.1.4. Types of Outcome Measures

Primary outcomes included: mean procedure time (min), mean stent modification time (min), 30-day survival rate (%), technical success rate (%), mean follow-up time (months), rates of complications (% or number of patients). As accessory outcomes, we collected data on technical details: mean cannulation time (min), fluoroscopy time (min), contrast agent volume (mL), and patient-associated characteristics: average intraoperative blood loss (mL), mean hospital stay duration (days), mean postoperative intensive care unit monitoring duration (days).

### 2.2. Search Methods for Identification of Studies

#### 2.2.1. Electronic Searches

The last electronic search was conducted in March 2024, utilizing the following databases: MEDLINE (PubMed), CAB Abstracts (Web of Science), CINAHL (Ebsco), Web of Science Core Collection (Web of Science), EMBASE, the Cochrane Library, ClinicalTrials.gov, Scopus Science Direct, Google Scholar. Searches included references without any restrictions on year and were limited to studies published in English. Retrieval experts from the medical library at the Poznan University of Medical Sciences consulted the searches. A complete list of research terms used for the present review is reported in Appendix A.

#### 2.2.2. Searching Other Resources

Reference lists of previous reviews and meta-analyses on FEVAR/TEVAR with 3D printing implementation were scanned. Gray literature, including dissertations, reports, and conference proceedings, were searched via OpenGrey.eu and PDF searches in Google.

### 2.3. Data Collection and Management

The search yielded 270 records after removal of duplicates (Figure 2). References were imported into the Rayyan software for screening by titles and abstracts. First, a random sample of 30 records was independently double-screened by one of the authors to assess inter-rater agreement. The inter-rater agreement was assessed using Cohen’s K was satisfactory, K  =  0.88. Any conflicts and questions related to the eligibility criteria were resolved by discussion with the co-authors before proceeding with the screening. After title and abstract screening, 92 references remained; full texts were screened for inclusion criteria. At this stage, two different authors (one being the first author, the rest being distributed among the remaining authors) independently screened a random sample of 80% of the records. The inter-rater agreement with Cohen’s K was good (K  =  0.82). Questions and conflicts were discussed and resolved among the authors, and doubts about the inclusion or exclusion of the remaining 20% of the papers were discussed for studies that needed further assessment and evaluation. One of the studies was excluded as it did not fulfill the inclusion criteria, but as it provided valuable information, we included a conclusion on it in Appendix A. 

Our extracted variables were: Reference, Source, Country, Number of patients, Type of surgery, Software, Model of the 3D printer, Polymer used for printing, Estimated cost of printing, Time spent for printing, Mean stent modification time, Name of the endograft modified, Sterilization technique, Mean time of cannulation, Fluoroscopy time, Contrast agent volume, Mean procedure time, Optimal angiographic result obtained, Average intraoperative blood loss, Mean hospital stay duration, Complications, Mean postoperative intensive care unit monitoring duration, 30-day survival rate, Mean follow-up, Types of settings and lists of them along with their respective units can be found in Appendix A.

#### 2.3.1. Assessment of Risk of Bias in Included Studies

Bias assessment was performed using the methodological index for non-randomized studies (MINORS) This tool for quality assessment was created for both comparative and non-comparative trials. For non-comparative trials, there are eight domains that can be graded 0 to 2 points, so a maximum of 16 points can be scored [26]. Point intervals were established as follows: 15–16—good quality, 9–14—moderate quality, ≤8—poor quality for non-comparative trials and 23–24—good quality, 15–22 moderate quality, ≤14—poor quality for comparative trials. Only data from studies graded “moderate” and “good” quality will be included in the final conclusion.

The assessment was carried out by two researchers, with any differences resolved through discussion. The consensus statement has been written on the templates available in Appendix A. 

#### 2.3.2. Measures of Treatment Effect

The treatment effect was measured as the mean operation and modification time as continuous data, along with the complications rate as the number of complications compared to the total number of treated patients for each study.

#### 2.3.3. Dealing with Missing Data

To tackle missing data issues, researchers searched for unpublished results in the browsers and websites described in method Section 2.2.1. Due to the novelty of the scientific area, no unpublished results were found to be eligible for review. To address missing data issues comprehensively, researchers attempted to retrieve unpublished results by contacting authors via email. Despite these efforts, no unpublished results were found eligible for review due to the novelty of the scientific area.

#### 2.3.4. Assessment of Heterogeneity

Heterogeneity between the study results has been assessed via Medcalc calculation for the random effect model. The heterogeneity between the studies was presented in a funnel plot graph.

#### 2.3.5. Assessment of Reporting Biases

The researchers have taken into consideration reporting bias. To address this issue, two researchers worked independently. Moreover, there was no selection of studies based on citation, location, or outcome.

## 3. Results

### 3.1. Description of Studies

The primary characteristics of our data sets are shown in Table 1 and Table 2. We compiled data from five publications, two from China, one from South Korea, and one from Poland and Germany each. The median patient cohort across the studies is 34.4 (range: 19 to 44). Two publications addressed TEVAR, one focused on FEVAR, and two on a combined approach with both FEVAR and BEVAR. The mean weighted stent modification time was 65.38 ± 10.59 min, which covers the range 37.63 ± 2.99 min representing the shortest and 109.6 ± 10.7 min as the longest time. The calculated weighted mean of procedure time from all studies is 249.95 ± 70.03 min with the longest procedure lasting 336 ± 72 min and the shortest being 147.84 ± 33.94 min. On average, optimal angiographic results were obtained in 97.21%, with four publications reporting 100% optimal angiographic results. The average 30-day survival rate was 97.6%, with four publications reporting a perfect 100% and one documenting a 30-day survival rate of 88%. Mean follow-up periods ranged from 6 to 16.14 ± 3.76 months. 

The study conducted by Fu et al. examined the effect of 3D guidance on fenestrated/branched thoracic endovascular repair (B/FEVAR) [27]. The research involves a group of 44 people (34 males and 10 females) with an average age of 59.84 ± 11.72 years. There were 22 acute and 8 long-term aortic dissections and 14 aneurysms treated with PMSGs modified on the 3D aortic model. A total of 132 branches were reconstructed and the average hospital stay was 9.91 ± 4.47 days. There was no need for open conversion [27].

Another study from China compared two groups: Three-dimensional (3D)-printing-assisted extracorporeal fenestration (*n* = 32) to conventional extracorporeal fenestration (*n* = 25) [28]. The patients were treated for Stanford Type B Aortic Dissection (TBAD). Here, 15 cases were classified as hyper-acute (<24h) and 42 as acute (1–14 days). The results indicated that there is a significantly shorter operation time for the 3D-assisted group (147.84 ± 33.94 min vs. 223.40 ± 65.93 min, *p* < 0.001). Also, there was a lower rate of immediate endoleak (3.1% vs. 24%, *p* = 0.048) which was also found to be significantly different, indicating better-tailored stent graft [28].

The study from Poland made by Rynio et al. was focused on juxtarenal and suprarenal aortic aneurysms, type IV thoraco-abdominal aneurysms, and type IA endoleak after EVAR [18]. A total of 162 fenestrations have been reinforced. The mean hospital stay was 8.06 ± 12.49 after endovascular treatment. There was a 12% (*n* = 5) 30-day mortality, due to acute kidney injury (*n* = 3), stroke (*n* = 1), and myocardial infarction (*n* = 1). The characteristics of treated patients involved: mean age 73.84 ± 6.99, hypertension in 95% of patients, and chronic kidney disease in 45% of cases. The authors of this study highlight the potential of 3D printing in PMSGs, especially in symptomatic cases where high mortality may be accepted [18].

The German study authored by Branzan et al. examined the effect of 3D-printed aortic models on PMSGs in thoraco-abdominal aortic aneurysm (TAAA) treatment [19]. Of the 19 patients who received the treatment, 13 were classified as having a symptomatic aortic aneurysm. The mean hospital stay was 17.3 ± 10.4 days. There was a 100% 30-day survival. There were only two reinterventions during the 14.4-month follow-up, with no aneurysm-related death observed. The treatment with 3D-guided PMSGs was found to be safe and feasible, especially in high-risk patients unsuitable for commercially available stent grafts [19].

A one-year study by Tong et al. treated 34 patients (19 thoraco-abdominal aortic dissections and 15 thoraco-abdominal aortic aneurysms) the mean age in the study group was 58 ± 14 years, and the mean hospital stay was 10.22 ± 3.65 days [29]. A total of 107 fenestrations were secured with 102 bridging stent grafts. There was one death reported after one week of surgery due to retrograde dissection rupture. The researchers concluded that the method of 3D-assisted stent-graft modification can be useful for complicated lesions involving crucial branches, however, more research was found to be needed [29].

Various software tools were utilized for 3D projects, with Mimics and Geomagic being the most common. Three publications described the usage of Eden260VS and two used Form 2 as their model of 3D printer. Multiple types of endografts were modified, with the Ankara covered stent and Endurant stent being the most common. Ethylene oxide was predominantly chosen as the sterilization method, as reported in four publications. Additionally, one publication documented the utilization of steam pressure and another hydrogen peroxide plasma (in addition to ethylene oxide).

There were 25 cases of early and late endoleak with the highest number being 12 reported in a study conducted by Rynio et al. (type I or III in 5 and II in 7 patients) and a minimum of 4 in Fu et al. [18,27]. Researchers reported 4 cases of infection among their patients (1—Branzan et al., 3—Zheng et al.), 5 neurological complications (1—Zheng et al., 2—Branzan et al., 1—Tong et al., Fu et al.), and 1 renal complications (Branzan et al.) [19,27,28,29]. Additionally, there were 2 cases of dissection (one in both Fu et al. and Tong et al.) and 2 cases of post-operative pain (Zheng et al.) [27,29]. A summary of the complications is viewable in Table 3.

### 3.2. Statistical Analysis

The means of modification time and operation time were analyzed as raw means using the meta-mean function (Figure 3 and Figure 4)

Meta-analysis was performed using the random-effect inverse variance model with Hartung–Knapp adjustment. Forest plots were used to present probabilities with 95% confidence intervals (95% CI) for individual studies and meta-analytic averages. Heterogeneity among studies was assessed using Cochran’s Q test using Heterogeneity Variance *τ*^2^ and Higgins & Thompson’s I^2^. Publication bias was assessed using funnel plots, Peters linear regression test was used to assess plot asymmetry (Figure 5 and Figure 6). Statistical analyses were performed using R software v4.3.2 (R Foundation for Statistical Computing, Vienna, Austria), and the meta (4.4-0) and meta (6.5-0) meta-analysis packages for R. *p*-values < 0.05 were considered significant. All tests were two-sided.

Statistical analysis revealed considerable heterogeneity among the included studies with the heterogeneity of I^2^ = 100%, Tau^2^ = 713.36 for modification time and I^2^ = 99%, Tau^2^ = 5301.01 indicating the substantial variability in effect sizes among the studies. Due to the magnitude of heterogeneity, further data synthesis was discontinued. The subgroup analysis was discussed in the discussion section.

### 3.3. Bias Assessment

Bias assessment was performed using the methodological index for non-randomized studies (MINORS). The assessment was performed by two researchers working independently. The studies’ quality has been measured using 8 domains for non-comparative trials and 12 for comparative trials. The overall effect of the assessment is displayed in Table 4. None of the examined studies was marked as “good quality”. The highest score among non-comparative trials (11) was given to Branzan et al. [19]. In comparative trials, it was only Zheng et al. with “moderate quality” (19 points) [28]. The research conducted by Tong et al. scored 11 points and was found to be of “moderate quality” [29]. Fu et al. together with Rynio et al. scored 9 points, the lowest among the included studies also receiving a “moderate quality” grade [18,27]. None of the given studies received points in D5 (Unbiased assessment of the study endpoint), and none of the studies reported blinding during the data collection process. The same issue was present in D8 (Prospective calculation of the study size), all the studies did not report the information about the study size. Also, domain D3 (Prospective collection of data) was reported inadequately for a majority of the studies except for that by Tong et al. [29]. This domain checks whether the protocol of data collection was established prior to the start of the study. One of the studies, conducted by Rynio et al., has reported a loss of more than 5% of the follow-up (D7), which resulted in no points in this section [18]. There was no research with an overall “poor quality” score. As mentioned above, the studies with “moderate quality” and “good quality” grades were included in the final conclusion.

## 4. Discussion

### 4.1. The Utility of 3D Printing in Urgent Surgeries

The use of 3D models for physician-modified stent grafts in elective surgeries poses the question of the utility of this method in life-threatening conditions. Some of the included studies tried implementing 3D printing in symptomatic aneurysm management. In one of the included studies, there were 13 symptomatic patients treated for TAAAs with no 30-day perioperative mortality and no aneurysm-related death in the 14.4-month follow-up, highlighting potential uses in urgent and complex aortic pathologies [19]. The study that compared conventional and 3D assisted PMSG also performed the surgeries on high-risk patients with hyper-acute symptoms (<24 h), and the results presented a significantly lower rate of immediate endoleak (3.1% vs. 24%, *p* = 0.048) in the 3D-assisted group [28]. Other parameters differed insignificantly between the two groups (*p* > 0.05), indicating the safety of the 3D procedure [28]. The study conducted by Rynio et al. presented the performance of nine surgeries using 3D-assisted PMSG in symptomatic patients, which equaled 20% of the treated patients [18]. It was also pinpointed that a custom stent graft from the manufacturer can sometimes take 12–15 weeks to be delivered, while the risk of rupture in giant aneurysms is high [18].

### 4.2. The Reference to Custom-Made Stent Grafts and Conventional PMSG

A study from 2018 involved 49 patients who underwent custom stent-graft implantation due to thoraco-abdominal aneurysm. The mean intervention duration was 330  ±  120 min [30]. That is higher than the majority of the studies included in this systematic review, except Tong et al. (336 ± 72min) [29]. Taking into consideration the fluoroscopy time (79  ±  34 min), it was also higher than some of the trials, e.g., Branzan et al. (55 min ranged 17–99 min). There was also a difference in the volumes administered at 212 ± 93 mL (Lucatelli et al.) vs. 77.7 ± 34.9 mL (Branzan et al.) [19,30]. It is noteworthy to highlight that those times and volumes in custom graft trials can be higher due to the complexity of the repair and it can also be the reason for higher 30-day mortality (10.2%) [30]. A systematic review from the year 2020 showed the 20-year experience of conventional physician-modified stent grafts. the results included the 30-day mortality, ranging from 0% to 8%, and a 14% incidence of type 3 or type 1 endoleak at the 14.8-month follow-up. It showed similar values to our systematic review when it comes to the spectrum of complications [31].

### 4.3. Utilization in Training

The integration of 3D models and virtual simulations into surgical planning has emerged as a promising approach in preoperative evaluation systems [32,33].

The printing success of such 3D models reached around 85% for standard clear resin in a study by Kaufmann et al. [34]. A study by Discher et al. found that the results of the surgeries were comparable whether the procedures were performed by a specialist or by a trainee who has trained in simulation-based settings, assisted by the specialist [35].

However, the diversity in presurgical training and standardized data collection methods among the included studies indicates a need for more uniform training in those settings [18,33,34]. Standardized protocols for data collection and procedural execution are essential for ensuring consistency and reliability in research outputs. Implementing standardized training programs could optimize the quality of data collected and enhance the validity of research findings.

### 4.4. Stent-Graft Technique and Material

An innovative approach involving the manufacturing of a flexible patient-specific AAA phantom utilizing a lost-core casting technique, coupled with a Particle Image Velocimetry (PIV) setup using an affordable laser source and global shutter camera, presents an avenue for conducting Fluid-Structure Interaction (FSI) simulations to deepen our understanding of the flow field [36].

Notably, the use of 3D models and virtual simulations has been demonstrated to streamline the surgical planning process, thus saving valuable time for both surgeons and patients [37,38]. Moreover, it leads to better vital organ circulation preservation and security of the length of a proximal sealing zone [38].

An analysis of software tools utilized for 3D projects underscores the prevalence of common platforms such as Mimics and Geomagic [39]. This emphasizes the need for standardized methodologies and workflows, with potential exploration into automated workflows in future studies.

### 4.5. Standardization and Optimization of the Procedures

The variation in stent-graft modification times across studies highlights the importance of procedural efficiency and optimization. None of the available research describes the fenestration process, which can stand for different fenestration techniques and can explain the large heterogeneity of the modification time [39,40]. Further research is warranted to identify strategies for streamlining stent-graft modification processes and reducing variability in procedural duration. While literature comparing fenestrated or modified stent grafts to standard stent grafts exists, more comparisons between different methods of fenestration are necessary [41]. The mean modification times can potentially vary significantly due to several factors, including the complexity of procedures, operator experience and skill, the availability and quality of equipment and technology, patient-specific factors such as anatomy and pathology, differences in surgical workflow, and variability in research methodology [42,43]. More complex endovascular procedures, less experienced operators, outdated equipment, and patients with intricate anatomies or medical conditions may contribute to longer modification times [44]. Additionally, variations in surgical workflow and data collection methods across studies can further impact reported modification times. Further research is needed to identify factors contributing to procedural duration and implement targeted interventions to enhance efficiency, potentially including a comparative analysis between different centers.

Some centers are exploring more direct diagnostic approaches to patients with aortic lesions. For instance, in the Czech Republic, a model to assess vascular wall stress from regular CT using data from past open repairs (using mechanical and histological tests from patient specimens) has been developed [45].

In situ fenestration combined with in vitro pre-fenestration presents an intriguing option for treating multiple aortic aneurysms [46]. Further exploration of this technique could provide valuable insights into its efficacy and safety for broader clinical applications.

It is posited that 3D models may contribute to the diagnosis and treatment of emergency vascular situations, such as floating aortic thrombi resulting from occult aortic dissection, as elucidated by Wang et al. in a detailed case report [47].

### 4.6. Enhancing Patient Outcomes and Risk Assessment

A 100% survival rate was shown in the 30-day follow-up in a vast majority of the chosen research. Meta-analysis states that there is a significantly higher survival rate in the perioperative period (first 30 days after the surgery) in endovascular operations compared to open surgical repair, although there is a lack of information about comparisons between endovascular operations using 3D models and other methods [11]. Future research should focus on this topic.

Further research is needed to evaluate optimal follow-up protocols and assess the durability of treatment effects over extended time frames. For example, a recent publication highlights the importance of EVAR management of follow-up using sac regression, which might be observed in duplex ultrasound scanning (DUS) or CT-angiography in 3-month, 1-year, 2-year, and 5-year follow-ups [48].

Although the research primarily focuses on classic EVAR surgery without 3D modeling, implementing this follow-up plan could prove beneficial for this patient cohort as well.

Our research identified several common complications, including endoleak, infection, neurological (paraplegia, ischemic stroke, or (transient) spinal cord ischemia), and renal disorders (acute kidney injury, etc.), which showcase similarities with complications seen in traditional physician-modified stent grafts [31].

Comparative analysis between traditional operating room (OR) procedures and 3D modeling-enhanced EVAR reveals similarities in certain complications, particularly related to renal and neurological issues. However, in the chosen studies, there were no cardiac, gastrointestinal, metabolic, or hematological perioperative complications observed, which were prominent among patients undergoing OR procedures. The distribution of complications among 241 patients undergoing traditional operating room procedures is as follows: cardiac accounted for 16.6%, respiratory for 16.2%, renal for 14.1%, neurological for 6.2%, metabolic for 2.9%, hematological for 34.3%, gastrointestinal for 17.8%, vascular for 4.6%, sepsis for 13.3%, and shock for 8.3% [49].

An essential point of differentiation between traditional OR and 3D-modeled endovascular operations is evident in the latest research, indicating a 30-day survival rate of 97.6% in the latter compared to 94.08 ± 1.9% in the former. Notably, most studies reported a 30-day survival rate of 100%, with only one study showing a rate of 88% [50].

Utilizing deep convolutional neural network models represents a promising avenue for accurately evaluating patient risk groups, with observed sensitivities of 100%. While currently applied to assess complications in EVAR procedures without 3D models, integrating artificial intelligence into pre-, intra-, and post-operative phases could significantly mitigate major complications and risks [51].

Research taken into consideration in this systematic review was carried out in countries located in Europe and Asia. Considering their differences in health expenses (% of GDP for 2022: China—7.05%; Poland—6.7%; Germany—12.7%; South Korea—9.7%), the amount of complications is prominently similar [52,53,54,55]. There are more common percentages of patients with reported complications between countries with the smallest and the highest national health expenditures, which might show that the main indicator for good outcomes is, e.g., surgical experience.

Intra-operative imaging plays a crucial role in EVAR procedures, with indications that 3D-2D fusion techniques are comparable to 3D-3D methods, offering reduced contrast dosage and radiation exposure for patients [56].

The adoption of personalized medicine approaches holds promise in optimizing treatment strategies based on individual patient characteristics and anatomical considerations, necessitating adjustments in training and clinical practice for vascular surgeons and other healthcare professionals [57].

Encouraging findings from a prospective study demonstrate the noticeable benefits of integrating 3D printing into open repairs of extensive thoraco-abdominal aortic aneurysms (TAAAs) [58]. However, further research through larger-scale studies, encompassing both open and minimally invasive repair techniques, is warranted to confirm these findings. While existing evidence suggests favorable safety and efficacy outcomes associated with this patient-specific approach, the limited scope of current studies underscores the pressing need for more extensive research to evaluate long-term outcomes adequately.

The significant heterogeneity observed across the protocols utilized in the studies has precluded the feasibility of conducting a comprehensive meta-analysis [59]. Despite all studies being single-arm longitudinal cohort studies without comparators, the analysis of received data might offer insights for future studies in this scientific domain. The absence of randomized case-control research within this domain is notable. Case-control studies serve as a cornerstone in modern vascular medicine for assessing the efficacy and safety of different endovascular approaches and techniques [60]. The lack of such comparative analyses indicates a potential gap in the existing literature, highlighting the necessity of incorporating this research methodology in future investigations.

It is worth mentioning the aspect of non-atherosclerotic aortic arch pathologies (NA-AAPs). This includes the bovine arch, arteria lusoria, or common bicarotid trunk, which increases the risk of aortic dissection or cerebral ischemia. The long-term treatment effect, especially in younger patients, depends on the quality of implemented stent grafts [61]. The personalized approach of 3D printing might also be an indication of NA-AAPs increasing the precision of the endovascular procedures by tailoring the stent grafts to the specific needs of patient anatomical aortic variants.

The distribution of endovascular procedures among the included studies encompasses a variety of approaches, including TEVAR, FEVAR, BEVAR, and their combinations, all of which modify the native hemodynamics flow in patients. Further studies comparing the length of the stent graft-covered aorta area with patient outcomes are needed. Noteworthy is the absence of remarkable innovations in stent-graft materials and design, which would signify a breakthrough in the field and enhance patient outcomes [13].

## 5. Limitations

The manuscript may face several limitations that could impact the interpretation and applicability of its findings. The primary challenge arises from the limited availability of English-language literature on 3D printing in EVAR, BEVAR, and TEVAR, given its relatively recent emergence. Moreover, some selected articles lacked sufficient detail, hindering the depth and scope of data compilation for the manuscript. There is a potential risk of publication bias since the review predominantly includes published studies, potentially overlooking unpublished or negative results. Additionally, the limited pool of publications incorporated in the analysis underscores the necessity for expanded datasets to support a more exhaustive and substantiated comprehension of endovascular interventions employing 3D printing technology. These limitations highlight the need for future research to address these gaps and provide a stronger understanding of the role of 3D printing in endovascular surgery.

## 6. Conclusions

This systematic review highlights the significant potential that remains underdeveloped within the area of endovascular procedures, particularly concerning the integration of 3D models and simulated preparations. Although complications like endoleak and infection persist, 3D modeling appears to reduce certain issues compared to traditional procedures. The integration of technologies like deep convolutional neural networks enhances risk assessment and procedural efficiency. Despite the scarcity of published research and the moderate quality of included studies, some of them highlight the potential use of 3D PMSGs in both elective and urgent treatment of aortic lesions. As such, it is evident that further research, collaboration, and standardization efforts are imperative to fully realize the benefits of 3D technology in enhancing patient outcomes and advancing endovascular surgery.

## Figures and Tables

**Figure 1 jcm-13-02977-f001:**
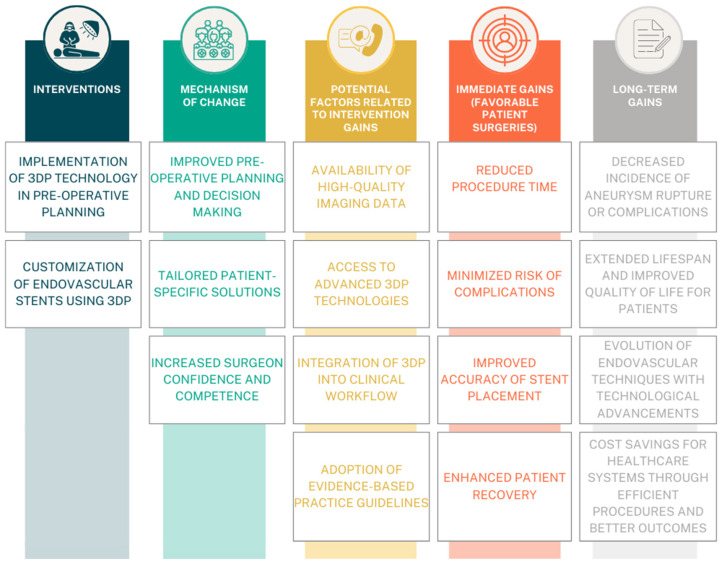
Illustration of a Theory of Change model for the intervention, focusing on the implementation of Three-Dimensional Printing (3DP) technology in pre-operative planning to enhance patient surgeries [20]. Immediate gains include reduced procedure time and decreased complications, facilitated by improved pre-operative planning and customized endovascular stents. Long-term gains involve enhanced recovery, cost savings, and the evolution of endovascular techniques through the continued integration of 3DP technology.

**Figure 2 jcm-13-02977-f002:**
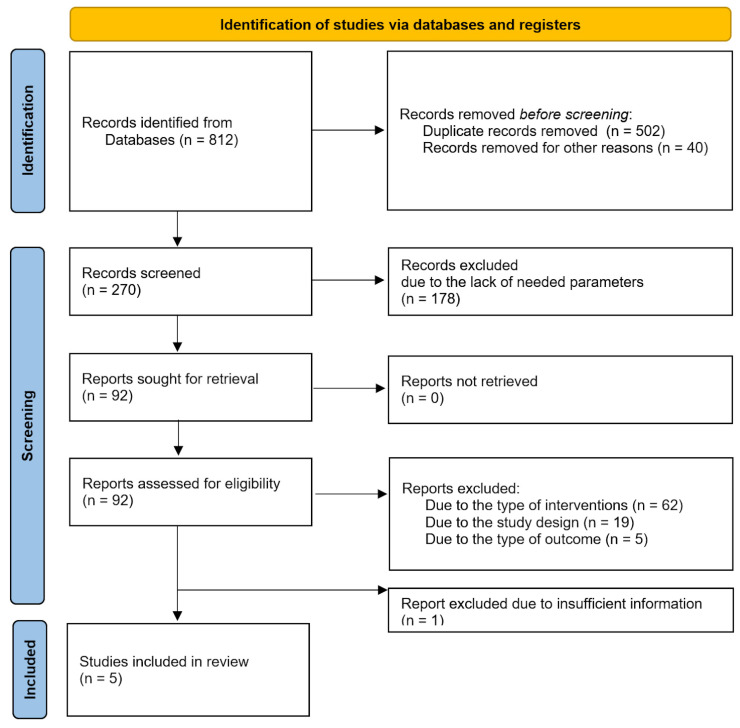
Flow diagram of the search and inclusion of references [22].

**Figure 3 jcm-13-02977-f003:**
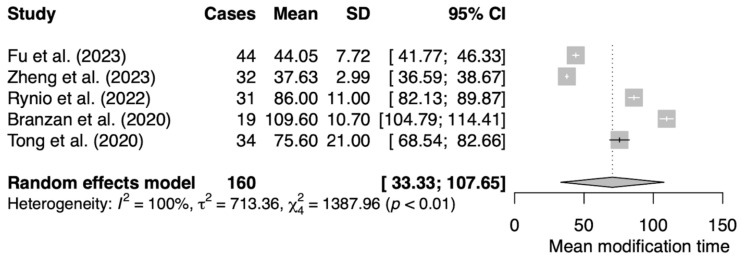
The random effect analysis for mean modification time [18,19,27,28,29].

**Figure 4 jcm-13-02977-f004:**
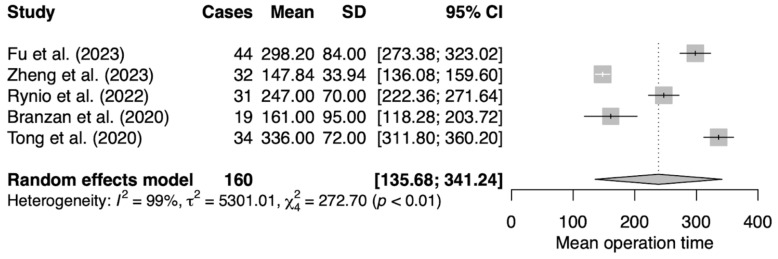
The random effect analysis for operation modification time [18,19,27,28,29].

**Figure 5 jcm-13-02977-f005:**
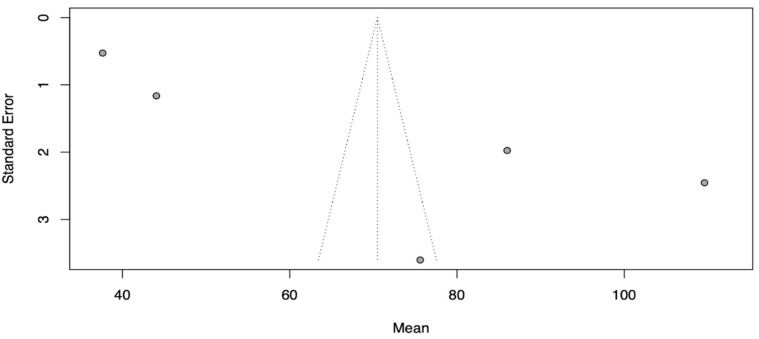
Heterogeneity funnel plot for modification time. The presented dashed lines (the funnel) indicate the region where 95% of studies would be expected to lie if there were no heterogeneity, grey dots indicate the study value.

**Figure 6 jcm-13-02977-f006:**
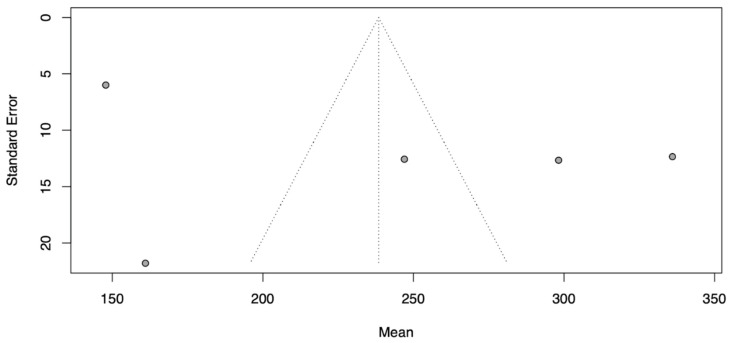
Heterogeneity funnel plot for operation time. The presented dashed lines (the funnel) indicate the region where 95% of studies would be expected to lie if there were no heterogeneity, grey dots indicate the study value.

**Table 1 jcm-13-02977-t001:** Overview of the surgical and treatment workflows in each of the included studies.

Author	Year of Publication	Patients (n)	Type of Surgery	Mean Modification Time [Minutes]	Mean Procedure Time [Minutes]	Optimal Angiographic Result Obtained (%)	30-Day Survival Rate (%)	Mean Follow-Up [Months]
Fu et al. [27]	2023	44	FEVAR, BEVAR	44.05 ± 7.72	298.2 ± 84	100	100	6 (42 patients,) 12 (35 Patients)
Zheng et al. [28]	2023	32	TEVAR	37.63 ± 2.99	147.84 ± 33.94	100	100	16.14 ± 3.76
Rynio et al. [18]	2022	43	FEVAR, BEVAR	86 ± 12	247 ± 70	86.05	88	14 ± 12
Branzan et al. [19]	2021	19	FEVAR	109.6 ± 10.7	161 ± 95	100	100	14.4
Tong et al. [29]	2020	34	TEVAR	75.6 ± 21	336 ± 72	100	100	8.5

**Table 2 jcm-13-02977-t002:** Overview of the 3D-printing approaches in each of the included studies.

Author	Software **	Model of the 3D Printer ***	Name of the Endograft Modified *	Sterilization Technique
Fu et al. [27]	Mimics, Geomagic Studio 2014, Geomagic Design Direct	Eden260VS	Ankura, Valiant Captivia, Endurant, Fluency, Viabahn	Ethylene Oxide
Zheng et al. [28]	Mimics, Geomagic Studio 2014	Eden260VS	Ankura	Ethylene Oxide
Rynio et al. [18]	3D Slicer 11.0, PreForm	Form 2	Valiant Captiva	Hydrogen Peroxide plasma, Ethylene Oxide
Branzan et al. [19]	Geomagic DesignX 2019	Form 2	Valiant Captivia, Endurant	Steam pressure
Tong et al. [29]	Mimics, Geomagic Studio 2014, EndoSize, CAD	Eden260VS	Ankura, Endurant, Zenith, Viabahn	Ethylene Oxide

* Endograft legend: Ankura—endograft by Lifetech Scientific Corporation, Shenzhen, China; Valiant Captiva—endograft by Medtronic, Dublin, Ireland; Endurant—endograft by Medtronic, Minneapolis, MN, USA; Fluency—endograft by BD/Bard, Tempe, AZ, USA; Viabahn—endograft by Gore Medical, Flagstaff, AZ, USA; Zenith—endograft by Cook Medical, Bloomington, IN, USA. ** Software legend: Geomagic Studio 2014 software (3D systems, Rock Hill, SC, USA); PreForm (Formlabs, Somerville, MA, USA); Mimics (version 21.0; Materialise, Leuven, Belgium); CAD (Autodesk, Inc, Mill Valley, CA, USA); EndoSize (Therenva SAS, Rennes, France). *** Printers legend: Eden260VS (Stratasys, Inc., Gilbert, AZ, USA), Form 2 printer (Formlabs, Somerville, MA, USA).

**Table 3 jcm-13-02977-t003:** Overview of the postsurgical complications in each of the included studies.

Author	Year	Country	Endoleak Type (Early and Late)				Infection	Neurological		Acute Kidney Failure	Retrograde Dissection	Post-Op Pain	All Patients
			I	II	III	Unknown		Cerebral Infarction	Spinal Cord Ischemia				
Fu et al. [27]	2023	China	2	-	2	-	-	1	-	-	1	-	44
Rynio et al. [18]	2022	Poland	3	7	2	1	-	-		-	-	-	43
Branzan et al. [19]	2021	Germany	2	1	-	-	1	-	2	1	-	-	19
Zheng et al. [28]	2023	China	4	1	-	-	3	1	-	-	-	2	32
Tong et al. [29]	2020	China	-	-	-	5	-	1	-	-	1	-	34

**Table 4 jcm-13-02977-t004:** Bias assessment using MINORS.

	D1	D2	D3	D4	D5	D6	D7	D8	D9	D10	D11	D12	Total
Fu et al. [27]	2	2	1	1	0	1	2	0	–	–	–	–	9
Zheng et al. [28]	2	2	1	2	0	2	2	0	2	2	2	2	19
Rynio et al. [18]	2	2	1	2	0	2	0	0	–	–	–	–	9
Branzan et al. [19]	2	2	1	2	0	2	2	0	–	–	–	–	11
Tong et al. [29]	1	2	2	2	0	1	2	0	–	–	–	–	10

D1—A clearly stated aim, D2—Inclusion of consecutive patients, D3—Prospective collection of data, D4—Endpoints appropriate to the aim of the study, D5—Unbiased assessment of the study endpoint, D6—Follow-up period appropriate to the aim of the study, D7—Loss to follow up less than 5%, D8—Prospective calculation of the study size, D9—An adequate control group, D10—Contemporary groups, D11—Baseline equivalence of groups, D12—Adequate statistical analyses. Scoring: 0—not reported, 1—reported but inadequately, 2—reported adequately. Overall score grading for non-comparative: ≥8—poor quality, 9–11—moderate quality. For comparative: ≥14—poor quality, 19—moderate quality.

## Data Availability

No new data were created or analyzed in this study. Data sharing is not applicable to this article.

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
