# Peer review of "The Utility of Three-Dimensional Printing in Physician-Modified Stent Grafts for Aortic Lesions Repair"

_jcm, 2024, doi:10.3390/jcm13102977_

Round 1

Reviewer 1 Report

Comments and Suggestions for Authors

3D printing is rapidly evolving and gaining traction in surgical practices. It holds particular promise for treating aneurysmal aortic lesions. By creating custom-designed fenestrated prostheses for each patient's unique anatomy, 3D printing could significantly reduce waiting times compared to traditional custom prostheses (which average 12-15 weeks, with a concerning 4% increase in mortality rate due to delays).

This review analyzed five recent (within the last five years) original research articles. While none were randomized controlled trials (RCTs) – as there are currently none in the literature – they examined average procedure and fenestration times, as well as complication rates. Despite a meticulous data analysis, the studies' significant heterogeneity in terms of surgical procedures, fenestration methods, times, materials, and software used prevented conclusive determination of the optimal strategy. Additionally, none of the studies detailed the fenestration technique.

Interestingly, the complication rates, including endoleak, were comparable to those observed in standard procedures where surgeons modify prostheses. However, the 3D-printed approach demonstrated a superior 30-day survival rate (97.6% vs. 94%) compared to open surgery. This suggests that 3D printing might be a viable alternative for patients with anatomies unsuitable for endovascular surgery.

The study also highlights the superiority of ethylene oxide for sterilization. The included tables and images are clear and informative.

While acknowledging its limitations, this review provides a valuable snapshot of the current state of 3D printing in surgery. It raises several intriguing possibilities, including the use of 3D printing for surgical training and as a potential alternative to open surgery for challenging anatomies. However, further research, particularly in the form of RCTs, is crucial to gather more robust scientific data and promote procedural standardization.

Questions for further investigation:

  • Are fenestration times, fluoroscopy times, and the amount of contrast medium used comparable between 3D-printed fenestration procedures, surgeon-modified fenestration procedures, and procedures using custom-made prostheses?
  • Do the potential benefits of shorter hospital stays and reduced need for intensive care outweigh the costs associated with 3D printing technology?

Reviewer 2 Report

Comments and Suggestions for Authors

I think this is an interesting study. The author decide to perform no further meta analyses after 2 analyses were with high heterogeneity. Perhaps a systematic review with description of all studies would be in place. 

11.      A noun should be added in line 32-33

22.      For female patients a diameter of 5.0 should be considered (line 41-42

33.      A reference should be added to line 43-45

44.      „mortality“ should be „perioperative mortality“, since long term mortality is comparable in EVAR and open surgery (line 50)

55.      Line 57-59 should be more clearly explained.

66.      A reference should be added to line 61-65. This reference should also support line 65-66.

77.      A reference should be added to line 91-93

88.      Why were TEVAR procedures also included (methods section line 116). In TEVAR no sidearms are necessary and the implantation is technically a lot easier as FEVAR or BEVAR.

99.      „time“ is missing in line 220

110.   What is meant with „optimal angiographic results“? (line 224)

111.   I think it is important to distinguish between type I and III endoleak and type 2 endoleak. Was this mentoined in the studies? (table 3)

112.   What is meant by „neurological“ (table 3)?

113.   What is meant by postoperative pain (table 3).

114.   What is meant by infection? Infection of the endograft? Table 3

115.   I don´t understand why no further analyses were performed? It could be that there was no significant heterogeneity for other outcomes.

116.   Can the ROBINS 1 tool also be used for single arm studies?

117.   If the author think a meta-analysis cannot be performed, perhaps a systematic review would be in place? The authors can describe the 5 published studies, including their specifics and technique.

Reviewer 3 Report

Comments and Suggestions for Authors

Thanks for submitting this interesting article evaluating utility of 3D printing in PMSG. After a deep evaluation of the manuscript some issues arise. The volume of the manuscript is too large and there is a risk that the aim of the study is not clear to the reader.

It is important to specify if usefulness of 3D printing of PMSG is considered in an elective or in an urgent setting. 

Several methods for PMSG preparation are described. It may be useful to compare with them.

Some sentence are unclear:

Line 50-52: "Advantages associated with EVAR encompass... lower migration rates". What is the meaning?

Line 60-61: "When using stent-grafts potential issues such... aortic sac regression are observed". What is the meaning?

Figure 1: Description must be improved.

Comments on the Quality of English Language

Several imprecisions were detected. Arguments are redundant. Rewriting is mandatory.

Round 2

Reviewer 2 Report

Comments and Suggestions for Authors

All the questions have been answered, I advise to accept the manuscript
